# [Re] VCNet: A Robust Approach to Blind Image Inpainting

**Furkan Kınlı**
Department of Computer Science
Özyeğin University
furkan.kinli@ozyegin.edu.tr

**Barış Özcan**
Department of Computer Science
Özyeğin University
baris.ozcan.10097@ozu.edu.tr

**Furkan Kıraç**
Department of Computer Science
Özyeğin University
furkan.kirac@ozyegin.edu.tr

## Reproducibility Summary

*In this study, we report and reproduce, on a large scale, the results of the article of a novel blind image inpainting architecture, namely VCNet, which jointly-controls the mask prediction and the blind inpainting modules. We have implemented this architecture from scratch in PyTorch, and then have conducted our experiments and blind inpainting evaluation on all datasets, as described in the article. We have achieved to reproduce the results qualitatively and quantitatively in most cases.*

**Scope of Reproducibility**

In the scope of this study, we validate the qualitative and quantitative results of VCNet on robust blind image inpainting.

**Methodology**

The original study achieves robust blind image inpainting by exploiting the mask prediction network (MPN) for guiding the blind inpainting branch (RIN). The paper has been implemented from scratch in PyTorch. We have conducted the synthetic data experiments on FFHQ and Places2 datasets, and also tried to achieve some of the blind inpainting evaluation tasks, as stated in the paper. Experiments have been completed on 1x RTX 2080 Ti in 4 to 6 days for each, and do not require any other significant resources, but GPU memory.

**Results**

We have achieved to reproduce the results qualitatively and quantitatively on a large scale. Qualitatively, MPN is able to learn the corrupted areas in input images in earlier steps reported in the paper, and RIN produces visually plausible outputs after extensive hyper-parameter tuning. Moreover, we measured the quantitative performance of the reproduced model with the metrics reported in the paper.

**What was easy**

The paper is well-written. The main components of VCNet, except Probabilistic Context Normalization, are composed of the common layers in PyTorch. Therefore, the general implementation is straightforward. We have no issues with implementing the loss functions, and all hyper-parameters are precisely indicated in the paper.

**What was difficult**

Due to the lack of computational resources, we could not fit the data with the batch size reported in the paper, and thus we need to adjust the learning rates and the number of training steps according to the appropriate batch size in our settings. GAN training is quite unstable while trying to tune hyper-parameters in the case of any change. Mask smoothing procedure and the input format for RIN are the issues that we fixed after communicating with the authors. For the experiments of face-swapping, we could not achieve to reproduce the reported results by applying exactly the same technique mentioned in the paper, *even after fine-tuning trials*. The failure cases can be found in this paper.

**Communication with original authors**

We were in contact with the authors since the beginning of the challenge. They swiftly answered our questions, and clarified some important missing points in training scheme and the architecture.

Preprint. Under review.

# 1 Introduction

The paper [1] proposes the versatile blind image inpainting pipeline that avoids the over-simplification of the assumptions in the previous blind inpainting studies. The previous approach for specifying the corrupted areas was to fill them with some constant values (*i.e.* black or white pixels) or simple data distributions [2,3,4,5,6,7]. However, such patterns, which are easily and almost perfectly identified by the network, can be considered as features to learn, instead of the contextual semantics. This may significantly degrade the overall performance of the network, especially when the corrupted area contains some unknown content. In the paper, this assumption is relaxed as the training data is generated by using similar natural images for filling the corrupted area in the original images with random strokes.

Secondly, the network architecture, namely *VCNet*, has two stages of mask prediction and inpainting with the guide of predicted masks representing the corrupted areas. This design has the capacity of inpainting the corrupted images without requiring any mask annotations during inference. To neutralize the negative effect of mask prediction errors on the generation part, VCNet utilizes the probabilistic context normalization (*PCN*), which mainly transfers the contextual information in different layers depending on the mask prediction probabilities. After training on well-known datasets, VCNet is able to inherently learn to inpaint the corrupted images in visually plausible and semantically coherent way, and it has slightly better or on-par quantitative results when compared to the other methods. Several different experiments for VCNet have been conducted to validate the blind inpainting results (*e.g.* face-swapping, different noise sources).

In this reproducibility report, we studied the new idea of generating training data and VCNet model architecture in detail, which contains implementing the architecture described in the paper, running the experiments, reporting the important details about certain issues encountered during reproducing, and comparing the obtained results with the ones reported in the original paper. Moreover, we discuss the further improvements for VCNet to enhance the quality of inpainting results.

# 2 Scope of reproducibility

The main contribution of the paper is to propose a robust blind image inpainting system which jointly trains a mask prediction module and an inpainting module. This system is also able to prevent the degradation of the performance of inpainting module due to the error propagation of mask prediction module by employing a new spatial normalization layer. To validate the qualitative and quantitative results of the paper, we mainly focus on answering the following questions: (1) Can we predict the noise mask by only using the corrupted images? (2) Can we validate the reported inpainting quality measured by the common inpainting metrics? (3) Can we achieve the qualitative performance demonstrated in the original paper? (4) What are the critical issues faced during the reproduction study, which may directly affect the performance? (5) Can we reproduce blind inpainting evaluation studies mentioned in the paper (e.g., different noise sources, raindrop removal and face-swapping)?

# 3 Methodology

Since the paper proposes a new model architecture, namely *VCNet*, we mainly focused on implementing this new architecture and its novel components within the details described in the paper. Moreover, we replicated the new training data generation mechanism where an image, as a noise source, fills the missing areas in the original corrupted image.

At this point, we found that the paper was well-written, and contains the most of the details required to reproduce the results. Initially, we wrote the code from scratch in PyTorch [8] since the original repository was not publicly available. The code writing process was not as challenging as we anticipated, but we still had some missing points not included in the paper. After communicating with the authors, we had a chance to access the private repository of the paper, and checked these points accordingly. In this section, we introduce the implementation details of VCNet, the points in the paper which were important for reproduction, and our experimental setup.

## 3.1 VCNet

The corrupted image $I$ in this setting is formulated as

$$\mathbf{I} = \mathbf{O} \odot (\mathbf{1} - \mathbf{M}) + \mathbf{N} \odot \mathbf{M} \tag{1}$$

where $\mathbf{O} \in \mathbb{R}^{h \times w \times 3}$ is the original RGB image, which is sampled from FFHQ and Places2 datasets, $\mathbf{N} \in \mathbb{R}^{h \times w \times 3}$ is the noise source, which is correspondingly sampled from CelebA-HQ and ImageNet datasets, and $\mathbf{M} \in \mathbb{R}^{h \times w \times 1}$ is the

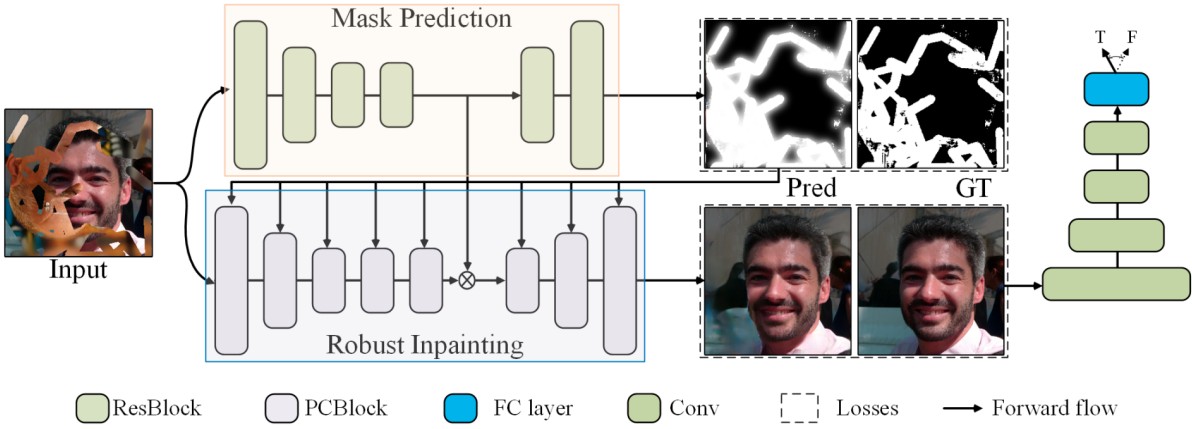

Figure 1: Overview of VCNet architecture. The input image is first fed into the Mask Prediction Network (MPN) to predict the noisy signals. Then, Robust Inpainting Network (RIN) produces visually plausible outputs without leveraging any mask annotations, yet utilizing the binary mask predicted by MPN. Obtained from the paper [1].

randomly-generated binary mask composing $\mathbf{O}$ and $\mathbf{N}$. The strategy of using real-world image patches for generating the noise source enforces the inpainting model to focus on the contextual information in the uncorrupted parts of images. This favorably alters the inpainting process in such ways that (1) the model generalizes better in real-world scenarios, and (2) the prior introduced by free-form strokes is moderated. Moreover, directly blending the ground truth $\mathbf{O}$ and the binary mask $\mathbf{M}$ leads to an issue for stable training, as we experienced and also mentioned in the paper. Therefore, we have to smooth the contact regions between $\mathbf{O}$ and $\mathbf{M}$ by applying iterative Gaussian smoothing ($k = 15, \sigma = 4$) to $\mathbf{M}$ in order to overcome this issue.

The proposed architecture for robust blind image inpainting, namely VCNet, is composed of two components introduced as Mask Prediction Network (MPN) and Robust Inpainting Network (RIN) (see Figure 1). The former predicts the noisy signals, (*i.e.*, binary masks) in the corrupted images, while the latter is responsible for inpainting the corrupted areas by utilizing the predicted noisy signals. During training, both networks are jointly trained after a certain number of steps. By doing so, RIN is able to locate the corrupted areas without any annotated supervision, and MPN is regularized by RIN due to the effort of preserving the contextual information.

MPN is composed of an encoder-decoder structure with residual blocks. The objective function is designed in self-adaptive manner where the effect of being positive or negative is adjusted by the densities in the output, and it can be seen as follows:

$$\mathcal{L}_m(\mathbf{M}, \hat{\mathbf{M}}) = -\tau \sum_p \mathbf{M}_p \cdot log(\hat{\mathbf{M}}_p) - (1 - \tau) \sum_p (1 - \mathbf{M}_p) \cdot log(1 - \hat{\mathbf{M}}_p) \tag{2}$$

where $\hat{\mathbf{M}}$ is the predicted binary mask, $p$ is the number of pixels, and $\tau$ represents the density of uncorrupted pixels in the ground truth mask. Note that although the objective function leads MPN to detect all corrupted parts, after starting jointly training with RIN, it eventually learns to ignore some corrupted parts, consistent with the contextual information.

Similar to MPN, RIN is composed of an encoder-decoder structure with probabilistic contextual blocks (PCB), which is a residual block variant spatially-guided by the predicted binary mask. This block comes up with an advanced solution to propagate the contextual information throughout the layers. Probabilistic context normalization (PCN) module appeared in these blocks transfers the contextual information in different layers for the corrupted areas without ignoring the features in the uncorrupted areas. The formula of PCN can be seen as follows:

$$PCN(\mathbf{X}, \mathbf{H}) = \beta \cdot T(\mathbf{X}, \mathbf{H}) \odot \mathbf{H} + (1 - \beta) \cdot \mathbf{X} \odot \mathbf{H} + \mathbf{X} \odot \hat{\mathbf{H}} \tag{3}$$

where $T(\cdot)$ is responsible for transferring the internal statistics of the instances, and formulated as follows

$$T(\mathbf{X}, \mathbf{H}) = \frac{\mathbf{X_P} - \mu(\mathbf{X_P}, \mathbf{H})}{\sigma(\mathbf{X_P}, \mathbf{H})} \cdot \sigma(\mathbf{X_Q}, (1 - \mathbf{H})) + \mu(\mathbf{X_Q}, (1 - \mathbf{H})) \tag{4}$$

Table 1: Hyper-parameters used in our experiments. Due to the computational constraint, batch size and learning rate in joint training differ from the original settings. We could fit the data to 1x RTX 2080Ti GPU with batch size of 4, and then adjust the learning rate and the number of steps accordingly.

| Hyper-parameters | | | | | | | | | |
|---|---|---|---|---|---|---|---|---|---|
| Data | | MPN | | RIN | | Joint Training | | Loss | |
| *Batch size* | 4 | *Base # Ch.* | 64 | *Base # Ch.* | 32 | *Optimizer* | ADAM | $\lambda_{mask}$ | 1 |
| *Image size* | 256 | *Neck # Ch.* | 128 | *Neck # Ch.* | 128 | *LR* | 2e−4 | $\lambda_{recon}$ | 1.4 |
| *Shuffle* | True | *Optimizer* | ADAM | *Optimizer* | ADAM | *Disc. LR* | 1e−3 | $\lambda_{sem}$ | 1e−4 |
| *Scaling* | True | *LR* | 1e−3 | *LR* | 1e−4 | *Betas* | (0.5, 0.9) | $\lambda_{tex}$ | 1e−3 |
| | | *Betas* | (0.5, 0.9) | *Betas* | (0.5, 0.9) | *Coeffs* | (2, 1) | $\lambda_{adv}$ | 1e−3 |
| | | *# Steps* | 20000 | *# Steps* | 20000 | *# Steps* | 200000 | $\lambda_{gp}$ | 10 |
| | | | | *Embrace* | True | *# Critics* | 5 | | |

where $\mathbf{X}$ is the input for PCN, $\mathbf{H}$ is the down-sampled binary mask, $\mathbf{X_P} = \mathbf{X} \cdot \mathbf{H}$, $\mathbf{X_Q} = \mathbf{X} \cdot (\mathbf{1} - \mathbf{H})$, and $\mu(\cdot, \cdot)$ is the weighted average and $\sigma(\cdot, \cdot)$ is the standard deviation of the composite feature maps governed by the binary mask. The objective function of RIN is the weighted combination of pixel-wise reconstruction loss, semantic consistency loss, texture consistency loss and adversarial loss

$$\mathcal{L}_g(\mathbf{O}, \hat{\mathbf{O}}) = \lambda_{recon}||\mathbf{O} - \hat{\mathbf{O}}||_1 + \lambda_{sem}||V_{\mathbf{O}}^l - V_{\hat{\mathbf{O}}}^l||_1 + \lambda_{tex}\mathcal{L}_{mrf}(\mathbf{O}, \hat{\mathbf{O}}) + \lambda_{adv}\mathcal{L}_{adv}(\mathbf{O}, \hat{\mathbf{O}}) \tag{5}$$

where $V_{\mathbf{O}}^l$ and $V_{\hat{\mathbf{O}}}^l$ represent the feature maps of the ground truth $\mathbf{O}$ and the output $\hat{\mathbf{O}}$ at level $l$, extracted from pre-trained VGG-19 network $V$. $\mathcal{L}_{mrf}$ is ID-MRF loss [9, 10], which computes the sum of the patch-wise difference between the generated content and its corresponding output patch, and enhances the details in the generated content. $\mathcal{L}_{adv}$ represents WGAN-GP loss [11, 12] for adversarial training. The coefficients for all components of the objective function are presented in Table 1.

## 3.2 Implementation Details

In our implementation, we have completely followed the instructions in the original paper. According to our observations, there are some critical points, which directly affects the training behavior and the overall performance of VCNet.

- All layers in VCNet are initialized with normal distribution with gaining of 0.02 (optional scaling factor).

- In the paper, the details about the discriminator is missing. According to the repository provided by the authors, discriminator is fundamentally designed as in SN-PatchGAN [17].

- Changing only the batch size to fit the data to the appropriate resources ruins the training behavior of the model. To overcome this, we adjusted the learning rate, and then the number of iterations required for the convergence has increased.

- Confidence-driven mask smoothing is a must to have a stable training. As mentioned in the original paper, applying smoothing driven by estimated mask confidence to the mask before feeding to PCN leads to minimize the mask error propagation in deeper layers and improve the contextual modeling. In practice, without this, the training diverges in earlier steps, and MPN eventually interprets any noise signals as a part of the main context (*i.e. predicts as all-zero binary mask*).

- During training on FFHQ, mixing the noise sources (*i.e. using CelebA-HQ and ImageNet together*) significantly improves the overall performance, and avoids any bias on facial dataset.

- Before feeding the images to RIN, we applied alpha blending to them with the binary mask predicted by MPN. This is represented with a parameter called *embrace* in the original code.

## 3.3 Hyperparameters

Since the main objective of this study is to achieve the reproduction of the original work qualitatively and quantitatively, we employed the default parameters reported in the paper, except batch size and learning rate, due to the computational constraint. At this point, we could not fit the data to our resources with the default batch size. After using the best possible batch size for our resources, we realized that the learning curve does not improve at all, or improve very slowly, during training with the default learning rate and our batch size. Therefore, we adjusted the learning rate and the number of steps accordingly with a number of trials.

### 3.4 Experimental setup

In this study, we have followed the same protocol described in the original paper for our all experiments. These experiments have been conducted in 1x RTX 2080Ti for approximately 4-6 days (*i.e. 4 days for FFHQ, 6 days for Places2, due to the more iterations required to converge*), and do not require any other significant resources, but GPU memory.

To generate the training data, we basically employed two dataloaders for the original images and noise sources, and blend them on-the-fly with the randomly-generated binary masks before feeding to the network. The parameters for generating random stroke brushes, as a binary mask, can be found in the configuration file in our GitHub repository. Our implementation and the trained weights are open-sourced, and can be found at `https://anonymous.4open. science/r/7164e419-4bc7-48a8-abdf-fd116596fed9/`.

### 3.5 Datasets

Following the paper, we have conducted our experiments on four well-known datasets: CelebA-HQ [13], FFHQ [14], Places2 [15] and ImageNet [16]. Due to the computational and time constraints, we picked 50K images from 10 classes (*auto showroom, bridge, canyon, house, mausoleum, ocean, park, rainforest, sky, street*) of Places2 dataset, and also employed the validation set of ImageNet for training. The image size is set to $256 \times 256$ where resizing is applied to FFHQ and CelebA-HQ, random cropping and padding is applied to ImageNet and Places2. We have configured our experimental setup with two settings where training images are picked from FFHQ and Places2, and corrupted by their corresponding noise sources (CelebA-HQ & ImageNet for FFHQ, only ImageNet for Places2).

### 3.6 Computational requirements

VCNet has *U-Net-like* structure with residual blocks in both MPN and RIN, and the components of its objective functions require to obtain some information from a *VGG-like* model. Considering this, training VCNet requires ~10GB GPU memory with the batch size of 4, while a single inference only requires ~1GB memory, and it took 225 milliseconds in RTX 2080Ti GPU.

## 4 Results

In this study, we mainly aim to reproduce the results in [1]. In general, we have achieved to reproduce quantitative and qualitative results of the experiments done on FFHQ and Places datasets. Following the paper, we measured the performance of our VCNet implementation by employing binary cross entropy (BCE) for mask predictions, peak noise-to-signal ratio (PSNR) and structural similarity index measurement (SSIM) for the inpainting quality.

### 4.1 Mask prediction by only using the corrupted images

As shown in Table 2, our VCNet implementation has on-par performance in mask prediction task on FFHQ dataset in terms of BCE, compared to the reported results in the paper. Also, it achieves superior performance on FFHQ dataset, compared to the most of the other methods mentioned in the paper, except Partial Convolutions. Even though the predicted masks are mostly near to be identical to the ground truth in the view of visualization (see Figure 3), there is still a gap between the performance of our implementation and the one reported in the original paper on Places2 dataset.

Table 2: Quantitative comparison of our implementation of VCNet

| Methods | FFHQ | | | Places2 | | |
|---|---|---|---|---|---|---|
| | BCE↓ | PSNR↑ | SSIM↑ | BCE↓ | PSNR↑ | SSIM↑ |
| *Contextual Attention* [18] | 1.297 | 16.56 | 0.5509 | 0.574 | 18.12 | 0.6018 |
| *GMC* [19] | 0.766 | 20.06 | 0.6675 | 0.312 | 20.38 | 0.6956 |
| *Partial Conv.* [20] | **0.400** | 20.19 | 0.6795 | 0.273 | 19.73 | 0.6682 |
| *Gated Conv.* [17] | 0.660 | 17.16 | 0.5915 | 0.504 | 18.42 | 0.6423 |
| *VCNet (original)* | **0.400** | 20.94 | **0.6999** | **0.253** | **20.54** | **0.6988** |
| *VCNet (ours)* | **0.439** | **24.76** | **0.7026** | 0.437 | **21.53** | **0.7070** |

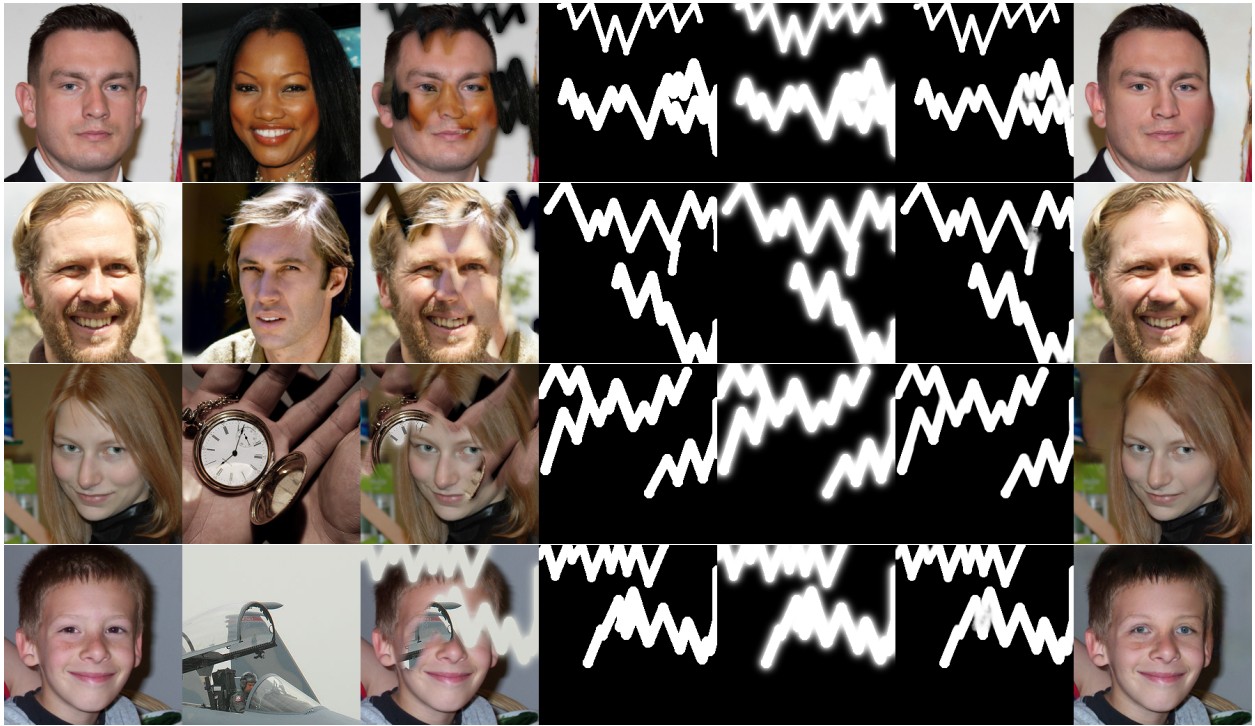

Figure 2: Qualitative results of our VCNet implementation on FFHQ dataset. Columns: (1) Original image from FFHQ dataset, (2) Noise source, (3) Corrupted image, (4) Ground truth binary mask, (5) The masks applied confidence-driven smoothing, (6) Predicted masks, (7) Inpainting output.

## 4.2 Inpainting performance

In this study, the main aim is to reproduce the inpainting results reported in the paper. Recall that we have conducted the experiments in such settings where the noise sources are taken from CelebA-HQ and ImageNet validation set for the original images from FFHQ, and ImageNet validation set for the original images from Places2. For both datasets, we were able to reproduce the results reported in the paper in terms of quantitative metrics for the inpainting quality (*PSNR, SSIM*). Particularly on FFHQ, we obtain better PSNR values (~4 dB) than the original results.

The main claim in the paper can be summarized as VCNet is able to produce visually more convincing and less degraded blind inpainting results since PCN resolves the problem of degradation in the output induced by the unknown contamination areas. In the scope of the reproducibility of the paper, our qualitative results show that it is possible to generate realistic inpainting results with the help of predicted binary masks, as plausible as the ones reported. In this regard, we can state that our qualitative results support this claim. Although the corrupted images may contain similar context with different priors, or even dissimilar context, VCNet achieves, first, to predict the binary mask of noisy signals from these images, and then inpainting the corrupted areas without any degrading yielded by the noise sources. The results of our VCNet implementation are shown in Figure 2 (*FFHQ*) and Figure 3 (*Places2*).

## 4.3 Blind inpainting evaluation studies: different noise sources, raindrop removal and face-swapping

One of the claims in the paper is that VCNet can deal with noise sources unseen during training, like *Gaussian noise* or *some constant colors*. To support this claim, and to show the effectiveness of new training data generation methodology followed in the training scheme of VCNet, the results for a corrupted image with different noise sources that are not included to the training are presented in the paper. Following the paper, we demonstrate the visual results of our implementation of VCNet model on FFHQ dataset corrupted by different noise sources in Figure 4. We can conclude that VCNet is robust to detect different noisy signals, *even the ones not included to the training*, and then to fill the missing parts in visually plausible way.

Secondly, the other claim in the paper is that the learned visual consistency ability on blind image inpainting task can be transferred into the other similar removal tasks with a few training data. To support this claim, raindrop removal task has been studied by employing VCNet with pre-trained weights trained on Places2 dataset. At this point, we re-designed

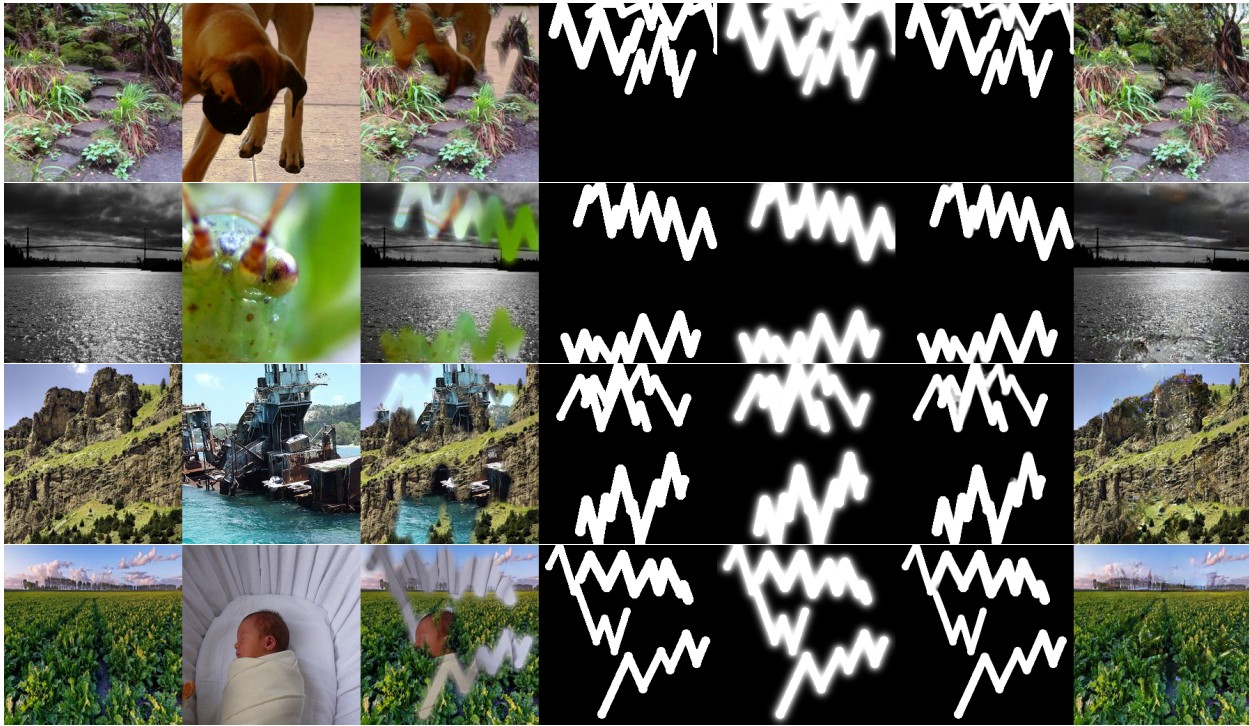

Figure 3: Qualitative results of our VCNet implementation on Places2 dataset. Columns: (1) Original image, (2) Noise source, (3) Corrupted image, (4) Ground truth binary mask, (5) Confidence-driven smoothing mask, (6) Predicted mask, (7) Inpainting output.

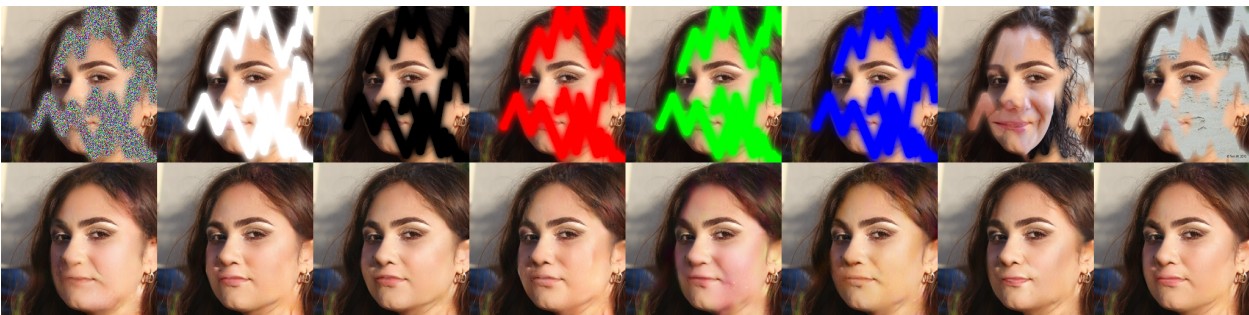

Figure 4: Visual results on FFHQ dataset corrupted by different noise sources.

our training setup to be compatible with raindrop removal model training (*i.e. the combination of blind image inpainting and image-to-image translation settings*), as described in the paper. We used the first 20 training images of the raindrop removal dataset, and fed them into VCNet with pre-trained weights for only 2,000 iterations. As shown in Figure 5, our implementation of VCNet has achieved to reproduce the raindrop removal task, so we can conclude that VCNet has the generalization property on the other similar tasks, *even by training a few instances for a small amount of time*.

Lastly, image blending on user-fed visual materials with the given image is the other potential application of this blind image inpainting system. The paper claims that VCNet can be used for filling the user-fed visual contents to edit the original images. We have tried to achieve the face-swapping task to support this claim, first as described in the paper. However, we could not execute the blind image inpainting process since MPN cannot detect any noisy signals in the corrupted images, and it considers all pixels pertain to the contextual information of the images. Thereafter, we changed the strategy of generating the corrupted images, and employed randomly-generated binary masks half-resized and center-pasted to the blank masks for being able to execute the second stage (*i.e.* RIN) of inpainting process. Despite all these attempts, we could not achieve to reproduce the results of the face-swapping task, and the failure cases can be seen in Figure 6.

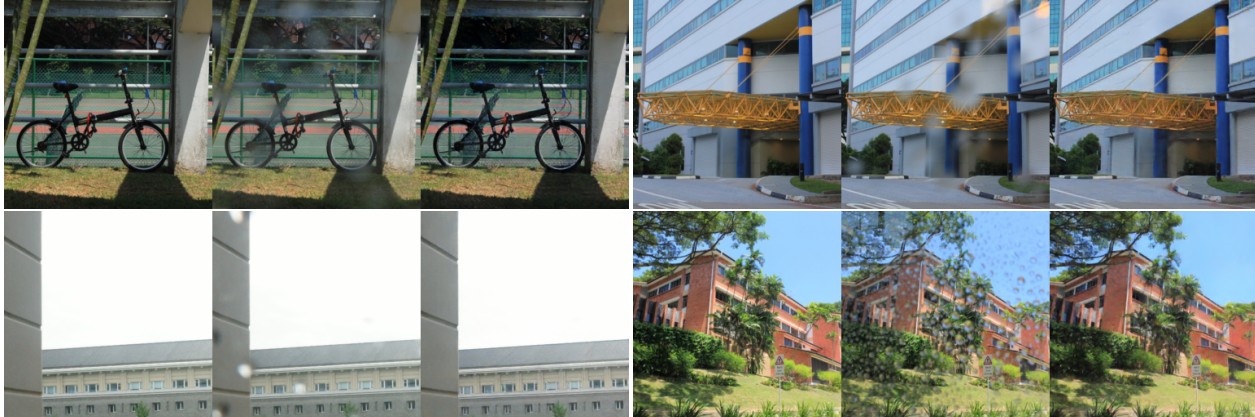

Figure 5: Visual results of the raindrop removal task. Columns: (1 & 4) Original image (2 & 5) Raindrop image (3 & 6) The output of our VCNet implementation.

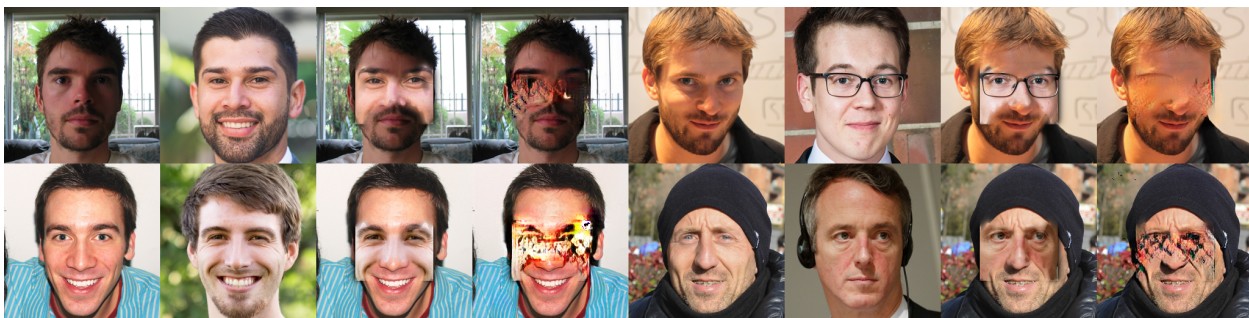

Figure 6: Failure cases in face-swapping task (*with our swapping technique*). By using the swapping technique applied in the paper, we could not apply any changes to the corrupted image, due to all-zero masks predicted by MPN. Columns: (1 & 5) Original image (2 & 6) Noise source (3 & 7) Corrupted image (4 & 8) Failures.

## 5   Discussion

We clearly state that the paper we reproduced was well-written. Even though there are a few missing implementation details (*e.g., the input format for RIN, the discriminator architecture and weights initialization procedure*), we have achieved to reproduce the visual results reported in the paper on a large scale. Quantitatively, our implementation of VCNet has on-par performances with the reported results for all metrics, and particularly, slightly higher PSNR value with the model trained on FFHQ dataset. Therefore, it is shown that the main claim and the most of the blind inpainting studies in the paper are supported by our experiments.

**What was easy:** The paper is well-written. The main components of VCNet, except Probabilistic Context Normalization, are composed of the common layers in PyTorch. Therefore, the general implementation is very straightforward. We have no issues on implementing the loss functions, and all hyper-parameters are precisely given in the paper.

**What was difficult:** Due to the lack of computational resources, we could not fit the data with the batch size reported in the paper, and thus we need to adjust the learning rates and the number of training steps according to the appropriate batch size in our settings. GAN training is quite unstable while trying to tune hyper-parameters if any change is required. Moreover, mask smoothing procedure and the input format for RIN are the issues that we fixed after communicating with the authors. For the experiments of face-swapping task, we could not achieve to reproduce the reported results by applying exactly the same technique, *even after fine-tuning trials*. The failure cases are shown in Figure 6.

**Communication with original authors:** We were in contact with the authors since the beginning of the challenge. They swiftly answered our questions, and clarified some important missing points in training scheme and the architecture. After a while, they have shared the private repository with us to make it possible to double check our code, and it was very helpful.

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
