# OpenReview forum: "[Re] VCNet: A Robust Approach to Blind Image Inpainting"
_ML_Reproducibility_Challenge/2020 — Reject_

### Official Review · AnonReviewer1 · 2021-02-25
**A reproducibility study of the VCNet blind image inpainting model**

**Rating:** 7
**Confidence:** 4

**Review:**

The given report examines the reproducibility of VCNet model for blind image inpainting of natural images. The model was proposed in the paper "VCNet: A Robust Approach to Blind Image Inpainting" by Wang et al. and it consists of two parts: a mask prediction network that tries to classify the input image into corrupted and non-corrupted regions and an inpainting network that performes the reconstruction. Furthermore a discriminator network is used for training with an adversarial loss.

**Reproducibility Summary:**
The report contains a reproducibility summary as required by the template.

**Scope of Reproducibility:**
The authors of the report clearly state the scope of their reproducibility experiments.

**Code:**
The model was implemented from scratch and the code was submitted. It is very well written but some doc-strings would highly improve the readability. Although the training weights were provided, I was not able to re-run the model without downloading the very large datasets. A small demo would have been useful to verify the implementation.

**Communication with Original Authors:**
The authors state that they were in contact with the original authors since the beginning of the challenge and were given access to the private repository to double-check their reproduced code.

**Hyperparameter Search:**
Most of the hyperparameters were set to the values reported in the original work are used. However, the authors decreased the batch size and adapted the learning rate due to compuatational limitation.

**Ablation Study:**
The ablation studies from the original work were reproduced. The authors succeeded to reproduce the raindrop removal experiments and the experiments using different noise sources, but fail to reproduce the face-swapping experiments. However, the authors spent a lot of effort to reproduce the face-swapping experiments.

**Discussion on Results:**
The results are discussed in an appropriate way. The report contains visual results as well as quantitative results using the metrics from the original work (binary cross-entropy-loss, PSNR, and SSIM). The authors are sincere that they were not able to reproduce the face-swapping experiments from the paper.

**Recommendations for Reproducibility:**
The authors point out that some implementation details are missing in the original paper (input format, discriminator architecture and weight initialization).

**Results Beyond the Paper:**
No results beyond the paper are reported.

**Overall Organization and Clarity:**
The description of the experiments is well-written and the report is overall well-organized. However, there are some issues in the section that describes the methodology, i.e., Section 3:
1. The authors use GT to denote the ground truth image which can be very miss-leading. I would recommend to use G or O (the latter one was used in the original work).
2. What is confidence-driven mask smoothing? The term is also not mentioned in the original work. A brief explanation would be useful.
3. In Equation 5: the variables V_{GT}^l and V_O^l are not defined. I suppose they are used to denote some feature maps in the l-th layer? However, this is just a guess.
4.The last two loss-terms L_{mrf} and L_{adv} are not further specified. The original work mentions that the WGAN-GP objective is used as adversarial term and the ID-MRF loss is used as texture consistency term. It would recommend to add these explanations to the report.

**Summary of Review:**
The authors did a great job to reproduce the results from the original work and I recommend to accept the paper to the ML Reproducibility Challenge 2020 after the issues in the above section have been adressed. I will improve my score as soon as these issues are fixed.

**Minor Remarks:**

- Equation (5): period should be placed after the equation (not in front of it)
- l 115: "with gaining of 0.02": I am not familiar with the term "gaining". Is it used to refer to the standard deviation/variance?
Figure 3 and 5: wrong wording: "(5) The masks applied confidence-driven smoothing"


**Familiar With The Original Paper:**

I have read the original paper

**Reproducibility Summary:**

Report has summary

---

### Official Review · AnonReviewer2 · 2021-03-06
**Useful Reproducibility study but with concerns**

**Rating:** 6
**Confidence:** 5

**Review:**

The paper seeks to reproduce the results of the paper titled “VCNet: A Robust Approach to Blind Image Inpainting”. Due to computational constraints, some hyperparameters like the batch size (and consequently others as well) were modified. For the same reason, only a subset of the original training data was used for training the models. Due to the above reasons, the quantitative results are somewhat different from the original paper. Despite such (statistically significant) differences, it seems reasonable to deduce that the core claims of the original paper are found to hold.

Overall, the reproducibility study is reasonable, and the changes necessitated by the lack of constraints do not substantially take away from the study conducted and reported herein. The study qualitatively substantiates the main claims of the original work and presents a novel ablation study that investigates the robustness aspect of the original model. This study will be useful to the audience of this challenge as well as the wider AI/ ML community interested in the original work.

I have some concerns regarding the stated scope and the alignment of the study design with the stated scope but perhaps it can be addressed in a revision. Similarly, the discussion on the results and the clarity should be improved. Overall, my recommendation is a borderline acceptance of the paper if the authors address the shortcomings listed below.
In the following, an evaluation of this paper on the metrics suggested by the RC 2020 challenge is presented:

Reproducibility Summary:

A 1-page summary is provided and adheres to the style guidelines. The major findings have been reported in the summary.

Scope of Reproducibility:

The authors have provided a brief and clear summarization of the problem statement and the proposed approach. However, more care should have been taken in stating the questions sought to be answered and aligning the study design to answering them.

(a) Q.1 – mask prediction – is evaluated. (b) Q.2 – quantitative results – instead should be inpainting quality as measured by PSNR and SSIM metrics. (c) Q.3 – whether it is possible to generate realistic inpainting results – is too vague. (d) Q.4 – whether PCN resolves the degrading inpainting performance – is not tested as an independent claim. (e) Q.5 – ablation studies mentioned in the paper – design choices (MPN, PCN ($\rho$), semantic consistency term, etc. is not tested. Instead, different use cases – generalization to unseen noise, raindrop removal, and, face swapping are investigated.

Code:

The code for the original paper is not publicly available. The authors have implemented the code from scratch in PyTorch. As mentioned in the report, the original paper contains detailed description of the architecture making the process easier but still some parts needed more clarifications.

Communication with original authors:

The authors mention that even though the original work is quite descriptive they had to reach out to the authors to clarify certain architectural details. The original authors gave access to the private repository and the concerning points were clarified.

Hyperparameter search:

The authors of the report had limited access to computational resources. All the experiments were conducted on a single RTX 2080Ti. Due to the limited GPU memory, the maximum batch size that could be used for training was 4, which is different from the original proposed work. To compensate for the smaller batch size, the authors have finetuned the learning rate and the number of iterations accordingly. This setting should be useful to audience with similar constraints on the availability of resources.

The report also highlights key details that were crucial to reproducing the results but were missing in the original paper, for example, weight initialization and the architecture details of the discriminator. All the combinations of hyperparameter settings used in the experiments are provided in Table 1 of the report.

Ablation study:

The authors claim to conduct an ablation study, but I am not sure if it can be called such. Ablation refers to studying the impact of modeling design choices on performance by carefully ablating (removing) parts of the model. Authors instead study the downstream used of the model – (a) unseen noise sources (including raindrops) during test, and, (b) the face swapping task. The first task is only demonstrated qualitatively (like in the original paper) while the authors are unable to replicate the results on the second task. However, neither is the second task properly defined in the original paper nor in this study.

In addition to not conducting any independent ablation studies, the authors did not even replicate the ablation studies conducted in the original work (MPN, for example).

Discussion on results:

The authors clearly state aspects that were easy/ difficult to reproduce and the steps they took to address the difficulties.

This work focused on reproducing the following results (different batch size and hyperparameters; smaller training data):

Mask Prediction: The quantitative results (BCE metric) on the FFHQ dataset are off (worse) by about 10% relative error while the results on the Places2 dataset are off (worse) by about 73%. Qualitative results (examples presented) show that the predicted masks are close to the ground truth masks. This is not reflected in the quantitative evaluation and is not adequately explained.

Inpainting quality: The results on the FFHQ and Places2 datasets on the PSNR and SSIM metrics have a relative error of (18%, 0.4%) and (4.8%, 1.2%). Since the SSIM metric is almost identical to the original (despite the differences in the training set up), the differences do not seem to be perceptually relevant, and it can be reasonably claimed that the performance from the original work was reproduced.

It is unclear if the above differences are due to insufficient hyperparameter tuning or due to the limited data used for training.

Recommendations for reproducibility:

none

Overall clarity and organization:

The report's overall clarity and structure is adequate. There are a few typos and awkward grammatical constructs that can be improved. The report should be carefully checked for such errors (e.g., typos in line 81 and 86; the mask is denoted using M not N).

Readability can be improved further by incorporating the more important equations in the methodology section.  In line 127 the authors claim that the original paper does not mention about alpha blending which is not true (refer page 5 of the original paper). The report should be carefully checked for syntactic error (typos in line 81 and 86; the mask is denoted using M not N).


**Familiar With The Original Paper:**

I have read the original paper

**Reproducibility Summary:**

Report has summary

---

### Decision · Program_Chairs · 2021-03-31

**Decision:**

Reject

**Comment:**

Overall reviews and/or the paper content not good enough for the AC to recommend to the journal.